*The Company of*
**Biologists**

# Epithelial fusion is mediated by a partial epithelial–mesenchymal transition

Varsha N. Tamilkumar[1,2], Harsha Purushothama[1] and Raj K. Ladher[1,*]

## ABSTRACT

Epithelial fusion is a fundamental morphogenetic process critical for the closure and compartmentalisation of developing organs. While widely studied in systems such as neural tube and palatal closure, the cellular transitions that enable fusion remain poorly understood. Here, we investigate epithelial fusion during chick otic vesicle closure and identify a transient population of cells at the epithelial interface that mediate this process. These otic epithelial edge (OE) cells exhibit distinct morphology, reduced apicobasal polarity, and dynamic junctional remodelling, including altered distribution of ZO-1, CDH1 and RAC1. Notably, OE cells lack basal contact and display high sphericity, consistent with a partial epithelial-to-mesenchymal transition (EMT) phenotype. Transcriptomic profiling of microdissected tissues reveals that OE cells constitute a transcriptionally distinct population, enriched for EMT regulators, extracellular matrix remodelling genes, and WNT pathway components. Among these, the transcription factors Grhl2 and Sp8 were specifically expressed at the OE and exhibited opposing roles in epithelial identity. CRISPR–Cas9-mediated knockdown of either gene led to disrupted CDH1 localisation, loss of OE cell morphology and failure in epithelial segregation. These results suggest that epithelial fusion requires a regulated, hybrid EMT state that balances junctional plasticity with tissue cohesion. Our findings demonstrate that fusion-competent epithelial cells are not merely passive participants but actively modulate their shape, polarity, adhesion and genetic identity to enable morphogenesis.

KEY WORDS: Epithelial fusion, Epithelial-to-mesenchymal transition (EMT), Morphogenesis

## INTRODUCTION

Epithelial fusion is a key morphogenetic process during organogenesis, enabling the shaping and compartmentalisation of developing tissues. During fusion, the edges of two epithelia are brought into close apposition, and through transient remodelling of junctional components, cells from opposing edges establish new contacts. In multiple embryonic contexts, this process converts what was once a continuous epithelial sheet into two distinct epithelia (Pai et al., 2012). Epithelial fusion is a conserved mechanism across species and occurs in a variety of developmental contexts, including *Drosophila* gastrulation and trachea formation, as well as vertebrate processes such as optic fissure closure, otic vesicle (OV) formation, body wall closure, palatogenesis, and neural tube closure (Alvarez and Navascués, 1990; Geelen and Langman, 1979; Gestri et al., 2018; Nikolopoulou et al., 2017). Failures in epithelial fusion are associated with several common birth defects, including coloboma, cleft palate, neural tube defects and omphalocele (Ray and Niswander, 2012).

Studies on fusion events such as optic fissure closure, palatal shelf fusion and neural tube closure have provided critical insights into the molecular and cellular processes that govern fusion. A recurring theme in these systems is the importance of the epithelial edge, the interface between two distinct epithelial domains. For instance, in neural tube closure, this is the boundary between neural and non-neural ectoderm; in palatogenesis, between the nasal and oral periderm; and in optic fissure closure, between the neural retina and retinal pigmented epithelium. Fusion can proceed from the apical surface, as in neural tube and palatal shelf closure, or from the basal side, as seen in optic fissure fusion (Chan et al., 2021; Lan et al., 2015; Nikolopoulou et al., 2017).

As epithelial edges converge, the leading-edge cells undergo significant remodelling to mediate fusion. In several systems, including wound healing, optic fissure closure, and *Drosophila* amnioserosa closure, these cells display a distinct character; they form actin-based protrusions, show reduced apical–basal polarity and altered intercellular junctions, and often lie adjacent to discontinuities in the basement membrane. These traits are suggestive of a transient loss of epithelial identity, a phenotype commonly referred to as a partial epithelial-to-mesenchymal transition (partial EMT) (Bahri et al., 2010; Futterman et al., 2011; Gestri et al., 2018; Nikolopoulou et al., 2017).

EMT is not a binary switch but a continuum of stable intermediary states with varying degrees of epithelial and mesenchymal features (Nieto et al., 2016). In the context of epithelial fusion, the moderation of epithelial traits does not always involve the acquisition of mesenchymal properties; often, it is simply the loss of some epithelial characteristics, such as tight junction integrity or apical–basal polarity (Bahri et al., 2010; Gestri et al., 2018; Nagai et al., 2022; Nikolopoulou et al., 2017). This partial EMT phenotype confers the flexibility required for leading-edge cells to undergo junctional remodelling and interact with the extra-cellular matrix (ECM), features essential for successful fusion. Interestingly, similar moderated epithelial states are observed in disease contexts such as cancer progression and fibrosis, where cells require plasticity to migrate or invade (Aiello et al., 2018; Cheung et al., 2013).

The inner ear is derived from the otic placode, a thickened epithelial region of the embryonic surface ectoderm. During development, the placode invaginates, bringing the border between the otic and non-otic ectoderm into close proximity. Once this contact is established, the OV pinches off from the overlying ectoderm and becomes internalised within the cranial mesenchyme (Ladher, 2017; Sai and Ladher, 2015). Fusion between the apposed epithelial edges facilitates this

[1]National Centre for Biological Sciences, Tata Institute for Fundamental Research, GKVK PO, Bellary Road, Bangalore, India 560065. [2]The University of Trans-Disciplinary Health Sciences and Technology (TDU), Attur Layout, Bangalore, India 560064.

*Author for correspondence (rajladher@ncbs.res.in)

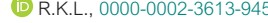 R.K.L., 0000-0002-3613-9456

Biology Open

segregation. Given the parallels with other epithelial fusion systems, we hypothesised that cells mediating OV fusion transiently adopt a less epithelial, more fluidic state that enables fusion and tissue segregation.

In this study, we characterise the cellular and molecular attributes of the otic epithelial edge (OE) cells involved in fusion. Morphologically, these cells are distinct from neighbouring epithelial populations, displaying altered shape and polarity. Through protein localisation and transcriptomic analyses, we identify features that mark OE cells as a specialised population within the continuous epithelial sheet. Furthermore, using CRISPR–Cas9-mediated mosaic knockdown of the transcription factors Grhl2 and Sp8, we show that perturbing the balance of epithelial and mesenchymal programs disrupts adherens junction dynamics and the characteristic cell shape changes required for fusion. These phenotypes were accompanied by a failure in epithelial segregation, suggesting that a regulated cell state transition, controlled by a network of transcription factors, is essential for OV fusion. Our transcriptomic profiling and *in-situ* hybridisation studies support the existence of this regulatory network and its role in shaping fusion-specific cell behaviour.

## RESULTS

### Cells in the OE are distinctly shaped

To characterise the cellular features of the OV during closure we first used scanning electron microscopy (SEM) to observe this process. *En face* views of the closing OV showed fusion starting from around HH16 and covering a period of around 8 h until HH17+, when the OV has just closed. To understand its cellular features, we imaged slices through the OV as it closes (Fig. 1A). Surface ectodermal (SE) cells showed a cuboidal morphology while OV cells were pseudostratified. At the border of these two domains, we detected cells that showed a more rounded phenotype and, in some cases, were detached from the basement membrane or below the apex of the epithelium (Fig. 1B,D and F).

To further understand and characterise the cell shape found at the OE, we used phalloidin to stain for cortical actin in thick sections of the closing OV. This allowed us to segment individual cells and assess their shape (Fig. 1C,E and G). The SE cells are cuboidal in shape and pseudostratified cells are elongated. The edge cells are more spherical and thus we evaluated the sphericity of cells in the OV to distinguish the three cell populations. The pseudostratified OV cells and the cuboidal SE cells are not as spherical as the edge cells. Cells with a sphericity index close to 1 were identified in the OE (Fig. 1H). To ask when these round cells first appeared, we assessed the different stages of OV closure. The rounded OE cells first appeared at HH16, 8 h before vesicle closure, and their numbers increased until vesicle fusion and the formation of the otocyst. By HH18, 2 h after fusion, round OE cells were not detected (Fig. 1I).

### OE cells show reduced apical–basal polarity

To understand the rounded edge cells further, we investigated the localisation of markers of epithelial polarity during closure. To quantify polarised expression, we developed a measure of localisation along the periphery of cells, the coefficient of variation (CV). Here, values close to zero show invariant intensity across the periphery, whereas values close to one show high variation. Zonula occludens-1 (ZO-1) is a tight junction component found at the extreme apical end of the lateral domain (Gumbiner, 1987; Stevenson et al., 1986). ZO-1 is restricted to the apical regions in both SE and OV cells (Fig. 2A, insets), with an average CV of around 0.68±0.66 and 0.72±0.15 respectively. In edge cells, however, ZO-1 is downregulated, with a CV of 0.26±0.067,

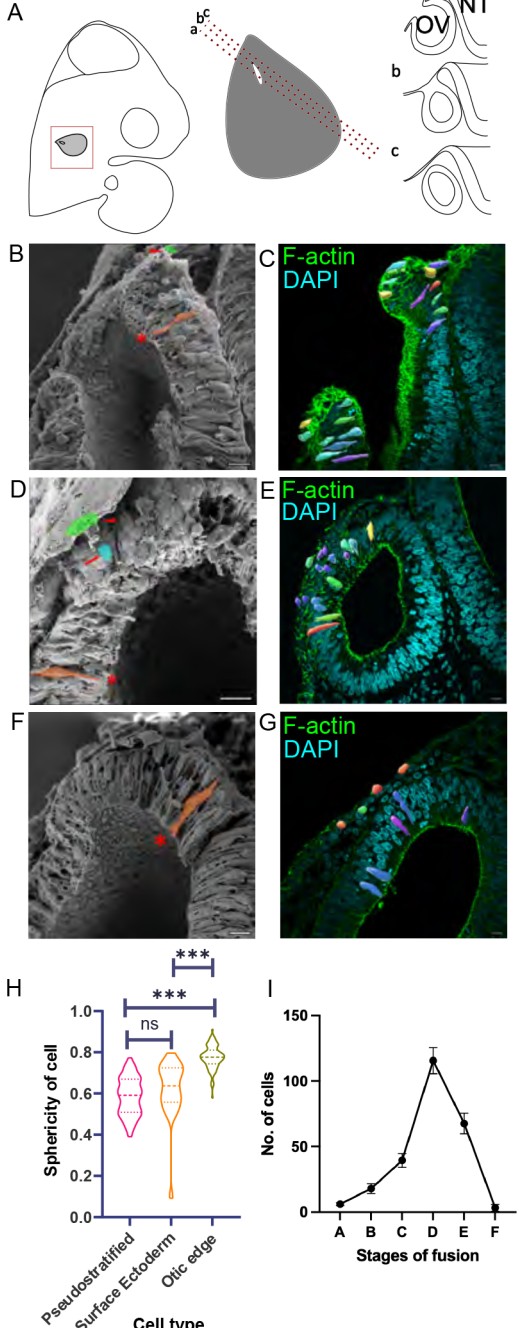

indicative of unpolarised localisation (Fig. 2B). The small GTPase Rac1 is known to regulate actin assembly and is localised to epithelia (Eaton et al., 1995; Ridley et al., 1992). Rac1 is apically localised in both SE and OV cells with a CV of 0.65±0.19 and 0.91 ±0.39, respectively (Fig. 2C insets and 2D). Previous studies had identified a role for Rac1 in filopodia formation in neural border cells prior to neural tube closure (Rolo et al., 2016). Consistent with this, SEM of similarly staged OV showed membrane protrusions bridging the anatomical gap between the two edges (Fig. 2H). We also observe Rac1-positive filopodia in OE cells as the edges are brought together (Fig. 2I). As the edges fuse, Rac1 is downregulated

**Fig. 1. Three cell populations in OV closure.** (A) Schematic for different planes of sectioning the chick OV. The sections exhibit different stages of epithelial fusion. (B,D,F) SEM of bisected OV across stages from HH15, HH17 and HH18 respectively. The pseudostratified OV cells are shaded in orange and marked by an asterisk. The cuboidal SE cells are shaded in green and marked by arrowheads. The round OE cells are shaded in blue and are marked by arrows. (C,E,G) Marking the shape of cells at different stages of epithelial fusion in OV sections. Sections stained with Phalloidin marked the cortical actin of cells, roughly approximating the cell boundary. This staining was used to mark the cell boundary across optical sections. The cells are colour-coded according to their sphericity, with blue marking a lower sphericity and red marking a higher sphericity. (i) When the edges are apart, (ii) when the edges are just meeting, (iii) when the edges are fusing and (iv) when OV closure is complete and has segregated from the SE. (H) One-way ANOVA of sphericity of cells. (I) Population of round cells in the otic edge across fusion stages. The X-axis denotes the following stages: A and B, HH15+ to HH16, when edges are apart; C, HH16+ to HH17 when edges are pushed closer and fusion is beginning; D and E, HH17+when fusion has occurred and remodelling has started; F, HH18, when the fusion is over and OV has segregated from the SE. Scale bars: 10 μm. For B,D and F number of cut embryos $n=5$ for each stage. For H, number of cells of pseudostratified $n=75$, OE=85 and SE=22. For I, number of sections $n=8$ for each stage.

in OE cells. Moreover, apical polarity is diminished and shows a CV of 0.36±0.1 in OE cells (Fig. 2D).

Cdh1 (E-cad) is a component of adherens junctions in epithelia (Takeichi, 2014; Yoshida-Noro et al., 1984). We find that it is localised to the lateral domain of SE and OV cells with a CV of 0.62 ±0.15 and 0.58±0.2, respectively (Fig. 2E insets and 2F). A population of cells in the edge show a loss of this lateral restriction, with a CV of 0.43±0.125. The radialisation of Cdh1 expression led us to probe further the epithelial properties of OE cells. Epithelia are characterised by their association with the basement membrane (Matlin et al., 2017). We thus asked if round cells in the OE were in contact with the basement membrane. We electroporated the OE with a low concentration of a plasmid encoding GFP. This filled some cells, enabling us to assess their shape and position. Using immunostaining for laminin, we found that the round cells do not contact the basement membrane. In contrast, SE and OV cells remain in contact with the basement membrane (Fig. 2G insets).

### Fusion generates interstitial cells between the SE and the OV

During the final stages of OV closure, the apposed epithelia establish contact and fuse to close the otic pore. Importantly, the two epithelia segregate so that the OV pinches off the SE. Soon after fusion and when remodelling has just begun, we noted a population of interstitial cells between the SE and OV (Fig. 3A-B). This population is transient, and no longer detectable after HH17+.

To ask if the interstitial cells underwent cell death, we investigated the activity of cleaved caspase 3, a marker of apoptosis (Porter and Janicke, 1999). Staining is observed at the junction of the SE and OV from HH16 (Fig. 3C left and middle panels). Later, we observe apoptosis in only some of the interstitial cells (Fig. 3C right panel). We have not observed cell migration from the OV edges at stages later than HH15. Laminin, a component of the ECM is localised contiguously across the SE–OV continuum. We did not see any breach in laminin localisation across stages from HH17-18 (Fig. 3D). This may suggest that interstitial cells have re-integrated into either the SE or OV.

The interstitial cells must re-establish their polarity after segregation. To investigate this process, we revisited the expression patterns of key polarity determinants - Rac1, and the ECM component

laminin. At early stages of OV development, laminin is expressed in a contiguous, robust pattern (Fig. 3D left panel). As fusion progresses, this expression becomes sparse (Fig. 3D middle panel), before returning to a robust localisation by the end of segregation (Fig. 3D right panel). Notably, we observe laminin punctae near some interstitial cells (Fig. 3E right panel, red arrow) that also exhibit polarised Rac1 expression, suggesting a role in orienting neighbouring cells (Fig. 3E Middle panel). Rac1 localisation in a single cell lamellipodia has been shown to play a role in collective movement of epithelial cells, without a need for establishing long range cell–cell junctions (Jain et al., 2020).

### Edge cells have a distinct transcriptomic profile

Altered junctional and polarity proteins appear to be critical for the neighbour exchange observed during epithelial fusion and the partial EMT phenotype, as reported in other systems (Aiello et al., 2018; Arnoux et al., 2008; Bahri et al., 2010; Huang et al., 2012; Shaw and Martin, 2016). This partial EMT state is thought to be maintained by a finely balanced expression of transcription factors promoting either epithelial maintenance or mesenchymal transition (Saitoh, 2023). Among these are Grhl2/3, Zeb2 and Snail2 (Nieto et al., 2016). We thus investigated their expression during OV fusion.

While Grhl2 and Zeb2 transcripts were detected at the site of fusion in HH17 embryos, Snail2 expression was restricted to the adjacent mesenchyme (Fig. 4A). Grhl2 expression persisted at the otic pore edges from HH14 through the completion of fusion (Fig. 4I). In contrast, Zeb2 expression was more transient and spatially restricted to the edge during the fusion stage (Fig. 4A iii). The co-expression of epithelial-stabilising Grhl2 and EMT-inducing Zeb2 suggests that cells at the fusion site may exist in a hybrid EMT state.

To explore this further, we performed bulk mRNA sequencing of micro-dissected tissues from HH17 embryos, isolating the OE, OV and SE (Fig. 4B). PCA and differential gene expression (DGE) analyses confirmed that the OE cells represent a transcriptionally distinct population from either OV or SE (Fig. 4C). Reactome analysis of DGE between the edge region (OE) and surrounding tissues (SE and OV) identified several enriched pathways, including gastrulation, WNT signalling, ECM remodelling and organogenesis (Tables S1 and S2), as summarised in the heatmaps (Fig. 4D). Some targets were then validated by whole-mount *in-situ* hybridisation (Fig. 4E–N), with expression domains predominantly confined to the OE, although occasionally extending into SE or OV.

Dach1 is a known inhibitor of EMT via Snail1 suppression in breast cancer (Zhao et al., 2015) and Six1 suppression in liver cancer (Cheng et al., 2018). Its expression was restricted to the OE and SE. Consistent with this reciprocal relationship of Dach1 and Six1, our DGE data revealed upregulation of Dach1 and downregulation of Six1 in OE versus OV (Fig. 4F). This corresponds with findings in later inner ear development, where knockdown of Dach1 in cochlear epithelium induces EMT and dysplasia (Miwa et al., 2019).

Several transcription factors implicated in developmental patterning and EMT were spatially restricted within the otic region. Dlx5/6 are known regulators of vestibular fate in OV and play a role in craniofacial patterning (Depew et al., 2002; Robledo and Lufkin, 2006). As previously observed (Groves and Bronner-Fraser, 2000), Dlx5 showed dorsal OV expression (Fig. 4G). Msx1 is known to drive apoptosis during limb development (Lallemand et al., 2009) and EMT during palate morphogenesis (Bendall and Abate-Shen, 2000; Levi et al., 2006). It is restricted to the OE (Fig. 4K), a region also undergoing apoptosis (Fig. 3C).

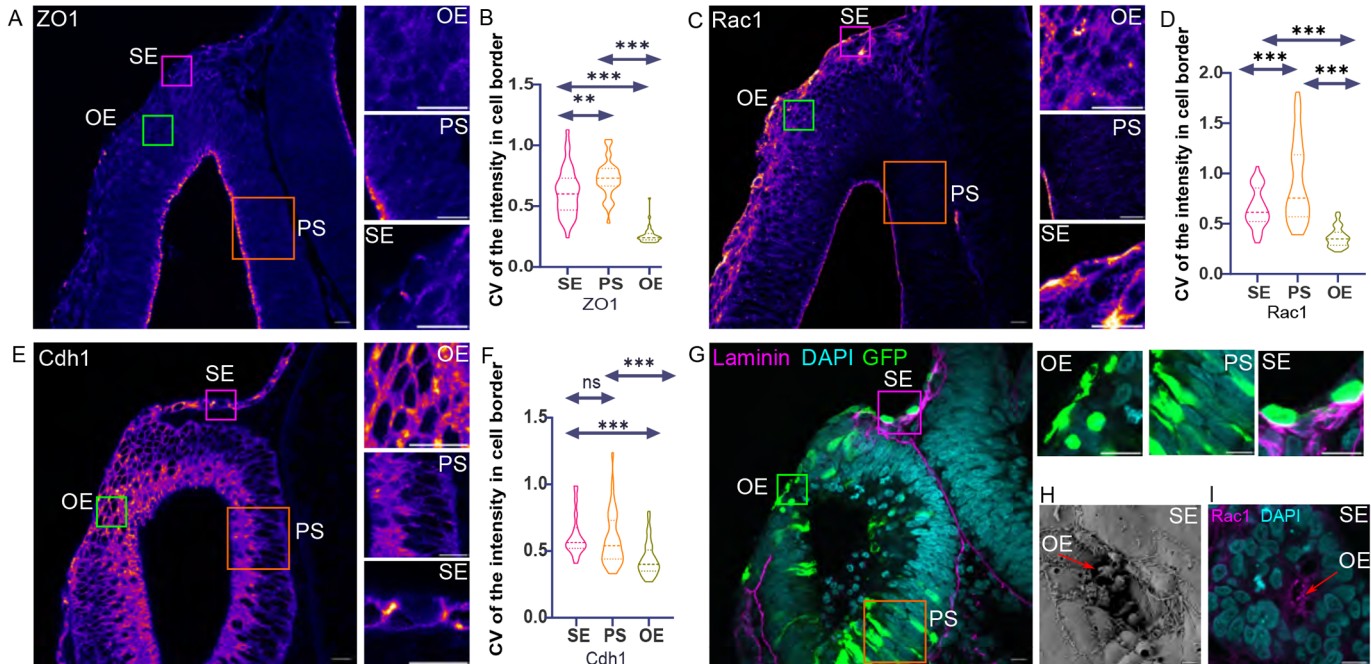

**Fig. 2. Cellular localisation of adhesion and polarity proteins at different stages of epithelial fusion.** (A,C,E) HH17+ staged embryo undergoing remodelling at the site of OV closure. The three insets are magnified and shown on the right. The insets represent the three cell populations: OE, round epithelial otic edge; PS, pseudostratified cells of OV; and SE, cuboidal surface ectodermal cells. Number of embryos for each protein is *n*=7. (B,D,F) Graphs depicting the coefficient of variation (CV) of fluorescence intensity of different proteins in three cell populations. The number of cells for each cell type in the graphs, *n*=40. Significance was tested using one way-ANOVA. (A) Tight junction protein ZO1. (B) CV of ZO1. (C) Apical polarity marker, Rac1. (D) CV of Rac1. (E) Adherens junction marker, Cdh1. (F) CV of Cdh1. (G) HH16 staged embryos electroporated with constitutively active eGFP and stained for laminin. The left panel shows a merged image of GFP positive cells in the OV. The inset shows nuclear staining of live cells at the site of fusion. The right panels are magnified views of the insets on the left. The red arrow marks a round cell expressing GFP and not attached to the basement membrane. Number of embryos, *n*=10. (H) SEM of closing vesicle showing cellular protrusions. Number of embryos *n*=5. (I) Whole-mount imaging of the closing OV, stained for Rac1, to visualise filopodia (marked by a red arrow) and nuclear staining. Number of embryos *n*=5. Scale bars: 10 μm.

Genes involved in signalling cross-talk and cell-ECM interaction were also differentially expressed at the fusion site. Epha4 mediates cell migration via TGFβ signalling in cancer (Hachim et al., 2017) had a transient, restricted expression in the OE (Fig. 4H). Itgb3 is a key integrin mediating ECM interaction and known for its role in metastasis (Kovacheva et al., 2021). It was expressed in the OE and the dorsal OV (Fig. 4J). Bambi, a negative regulator of TGFβ and BMP signalling induced by WNT signalling (Lin et al., 2008; Sekiya et al., 2004) was also transiently expressed at the OE (Fig. 4E).

Wnt3a is known to induce EMT (Qi et al., 2014), showed persistent expression in the OE even after fusion. This suggests additional roles in inner ear patterning, likely in dorsal–ventral patterning of the otocyst (Riccomagno et al., 2005) (Fig. 4N). The transcription factor SP8 acts downstream of Wnt3, Fgf10 and Bmpr1A during mouse development (Bell et al., 2003). It is also implicated in inner ear development in Xenopus (Chung et al., 2014) and EMT in hepatoblastoma (Wagner et al., 2020). It showed restricted OE expression (Fig. 4M). Rac3, a regulator of actin cytoskeleton dynamics and invasive behaviour in breast cancer (Gest et al., 2013), was expressed at the fusion site, further supporting a role in cell remodelling (Fig. 4L).

Overall, we observed a spatially restricted expression of transcription factors and signalling molecules that either promote or inhibit WNT, TGFβ, and BMP signalling at the site of fusion. This likely sustains the partial EMT state of OE cells. Notably, TGFβ signalling has been implicated in epithelial fusion in other developmental contexts (Iwata et al., 2011; Knickmeyer et al., 2018; Nakajima et al., 2000). While many of these genes are well-studied in cancer and adult tissues, their coordinated expression here

suggests a developmental program that mimics, but does not fully execute, EMT during embryonic epithelial fusion.

## Molecular perturbation of epithelial fusion

Based on their expression patterns and putative roles in epithelial dynamics and EMT regulation, Grhl2 and SP8 were selected for functional perturbation. Grhl2 is a key transcription factor that regulates epithelial integrity by controlling the expression of Cdh1 and Claudin4 (Werth et al., 2010). During embryogenesis, its role appears context-dependent; while some studies suggest it suppresses EMT in cancer cell lines, others highlight a more ambiguous involvement (Cieply et al., 2012; Jolly et al., 2016; Wang et al., 2023; Werner et al., 2013). Mouse models with Grhl2 knockouts exhibit neural tube (NT) and organ defects, varying with the specific allele (Pyrgaki et al., 2011). In zebrafish, Grhl2b mutants show inner ear phenotypes resembling human non-syndromic deafness DFNA28 (Han et al., 2011). Neural tube defects in mouse Grhl2 knockouts have been linked to Cdh1-dependent mechanisms (Nikolopoulou et al., 2019). We detected Grhl2 expression at the otic edge prior to fusion, and this expression was downregulated after segregation in the OE. SP8, a transcription factor involved in WNT signalling and limb and NT development, has also been implicated in inner ear formation (Chung et al., 2014).

To investigate the functional roles of Grhl2 and Sp8, we performed CRISPR–Cas9-mediated mosaic knockdowns (KDs) via electroporation. GFP-positive cells indicated successful uptake of CRISPR constructs (Fig. 5A). In embryos electroporated with control gRNAs, CDH1 expression appeared similar to wild-

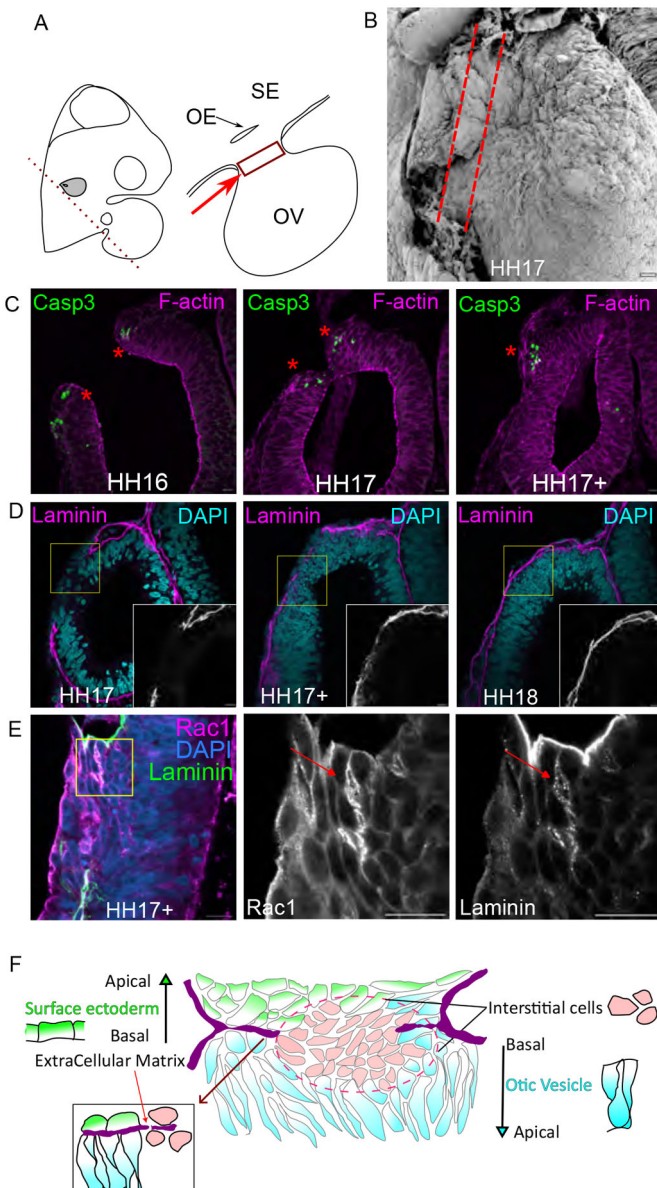

**Fig. 3. Interstitial cells aid in fusion and segregation.** (A) Schematic of HH17 chick embryo depicting the plane of slicing, to observe the OV. The red box and arrow show the location of interstitial cells. The SE is the topmost layer, which opens into the OV through the otic edge, as shown in the schematic. The SE and the OV are part of the same contiguous epithelium. (B) SEM of HH17 OV, the dashed red lines mark region below the SE and above the OV, where the interstitial cells are present, $n$=4 embryos. (C) Cleaved caspase3 marks cells undergoing apoptosis. The red asterisk marks caspase3 positive cells at the site of fusion, $n$=3 embryos for each stage. (D) Laminin is a basement membrane marker. A contiguous robust localisation of laminin is seen when the edges are fusing at HH17 (left panel). The localisation of laminin becomes sparse during remodelling, as marked by the red arrow at the site of fusion (middle panel). After the segregation of the epithelia, laminin is seen to be localised in the basement membrane of both the SE and the internalized OV (right panel), $n$=7 embryos for each stage. (E) Airyscan images of interstitial cells stained for Rac1 and laminin show a polarized expression and seem to have laminin punctae around them (red arrow), $n$=3 embryos.

type patterns (Fig. 5B). However, embryos electroporated with Grhl2-sgRNA displayed misregulated CDH1 expression at the fusion site. Aberrant CDH1 accumulation was detected within OE cells,

regardless of green fluorescent protein (GFP) positivity (Fig. 5C). Moreover, segregation between the OV and SE was incomplete (Fig. 5C). In contrast, SE cells that were GFP-positive exhibited complete loss of CDH1 (Fig. 5C), reminiscent of the switch to CDH2 seen in Grhl2-knockout mice (Nikolopoulou et al., 2019).

Sp8 mosaic knockdowns also resulted in ectopic CDH1 accumulation at the OE and loss of expression in the SE (Fig. 5D). Furthermore, the sphericity of cells in the OE and SE was significantly reduced compared to control embryos (Fig. 5E), suggesting altered mechanical properties or junctional remodelling. Although the CRISPR knockdowns were mosaic, the phenotypes observed were tissue-wide. This broader effect could be due to the mechanical and signalling integration properties of CDH1, which transmits cortical actin tension between neighbouring cells (Lecuit and Yap, 2015). Interestingly, the pseudostratified morphology of OV cells remained largely unperturbed. This is consistent with the absence of Grhl2 and Sp8 expression in the OV, suggesting their primary function is restricted to the OE and SE. These findings support a model in which Grhl2 and Sp8 contribute to cell shape regulation, CDH1 localisation, and ECM remodelling, all of which are essential for proper epithelial fusion and segregation.

## DISCUSSION

Epithelial fusion is a critical event in embryogenesis, with failures resulting in a range of developmental disorders such as neural tube defects, cleft palate, and body wall closure defects (Copp et al., 2015; Ray and Niswander, 2012). While significant work has focused on genetic contributions to fusion events, particularly in single-gene mutation models, these do not fully capture the complexity of fusion failure in human birth defects. Our study takes a complementary approach by investigating the dynamic cell behaviours, junctional remodelling, and ECM interactions that occur during OV fusion, using the chick embryo as a model system.

One of the key findings from our study is the identification of a transient population of round cells located at the boundary between the SE and OV. These cells display distinct morphological and molecular features that evolve across fusion stages. Immunostaining for CDH1, ZO1 and RAC1 revealed dynamic changes in their localisation (Fig. 2), indicating a process of junctional remodelling.

Epithelial remodelling in this context likely involves neighbour exchange and transient loss of stable cell–cell contacts, mediated through endocytic trafficking of adhesion proteins. Previous studies have shown junction remodelling being mediated by intracellular trafficking (Levayer and Lecuit, 2013; Roeth et al., 2009; Sigismund et al., 2021), often in response to mechano-chemical stimulus; for instance, in the *Drosophila* wing, p120-catenin is released under tension, promoting CDH1 endocytosis (Iyer et al., 2019). Similar tension-mediated mechanisms may operate during otic fusion, where tissue stress, arising from the buckling of the otic edge, may enhance CDH1 turnover. This process could be further regulated by Rac-family GTPases, which link CDH1 endocytosis with actin cytoskeleton remodelling (Baum and Georgiou, 2011) and is consistent with the observed expression of Rac1 and the specific expression of Rac3 in fusing cells.

The final phase of fusion is marked by ECM remodelling, particularly changes in laminin distribution (Fig. 3D,E,F). Initially robust and continuous, laminin becomes sparse as fusion proceeds. It is reassembled into distinct basement membranes as the epithelia segregate. This remodelling resembles epiboly during gastrulation, where tissue thinning occurs through radial intercalation of deeper cells into superficial layers (Keller and Hardin, 1987; Trinkaus and Lentz, 1967). A similar unpolarized intercalation is also likely

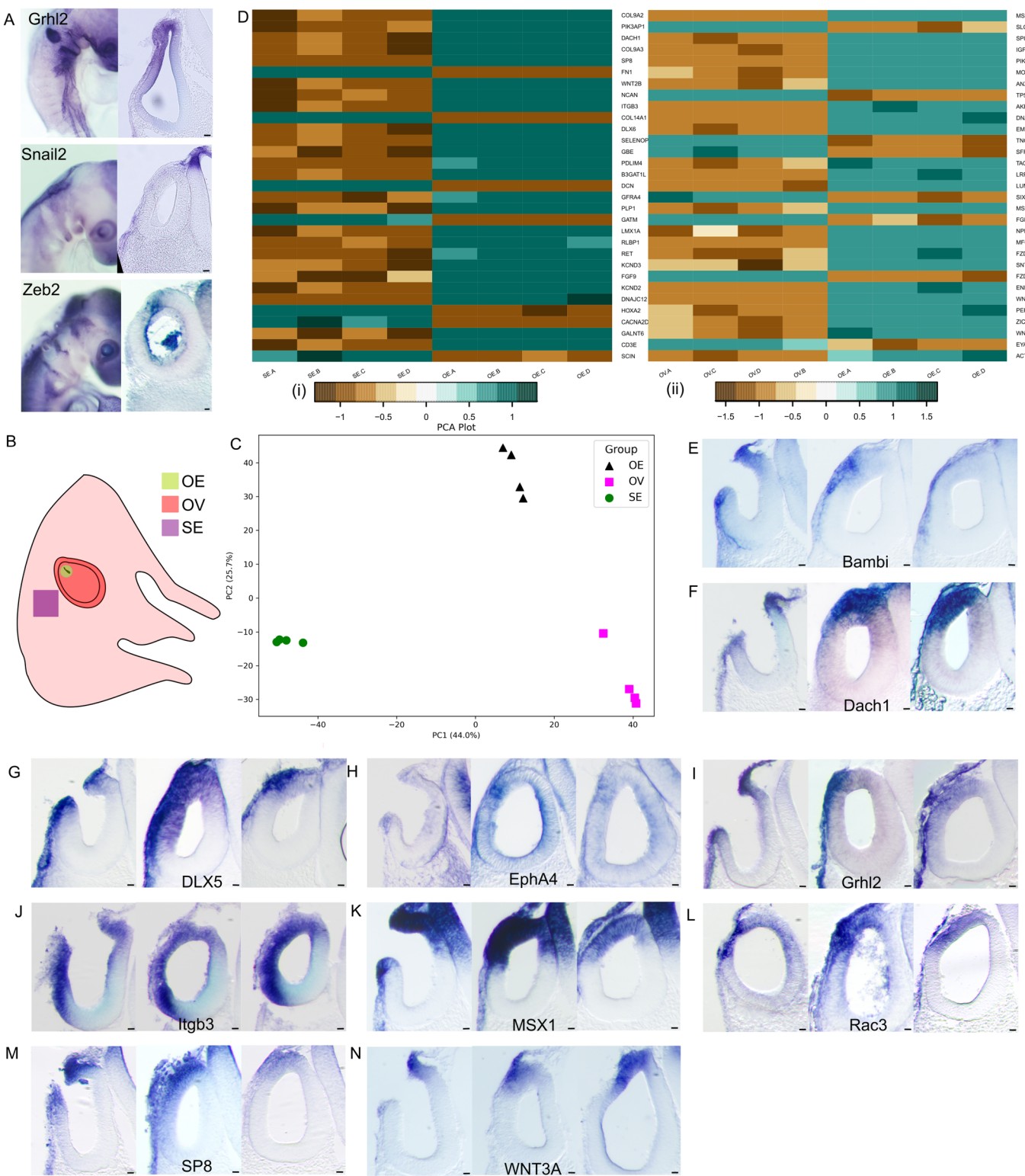

**Fig. 4. Transcriptomic state of the edge cells.** (A) Whole-mount and section of HH17 OV whole-mount *in-situ* hybridisation for the genes *Grhl2*, *Snail2* and *Zeb2*. *n*>3 embryos for the three genes. (B) Side view of HH17 chick embryo showing the three types of tissue taken for bulk-RNA sequencing: OV, SE and OE. The letters A-D denote the biological quadruplicates for each tissue type. (C) The multi-dimensional scaling (MDS) plot shows quadruplicates of the same tissue clustering together and the three tissue types segregated considerably from each other. This confirmed the quality of the sample collection. (D) Heatmap of top 30 differentially expressed genes (DEG) in OE with respect to OV and OE with respect to SE. (E-N) whole-mount *in-situ* hybridisation sections of HH17 embryos of some gene targets identified from RNA-seq data for three stages of epithelial fusion as mentioned in Fig. 2. The gene names are mentioned in the respective figure panel. Scale bars: 10 µm; *n*=4 embryos for each gene.

Biology Open

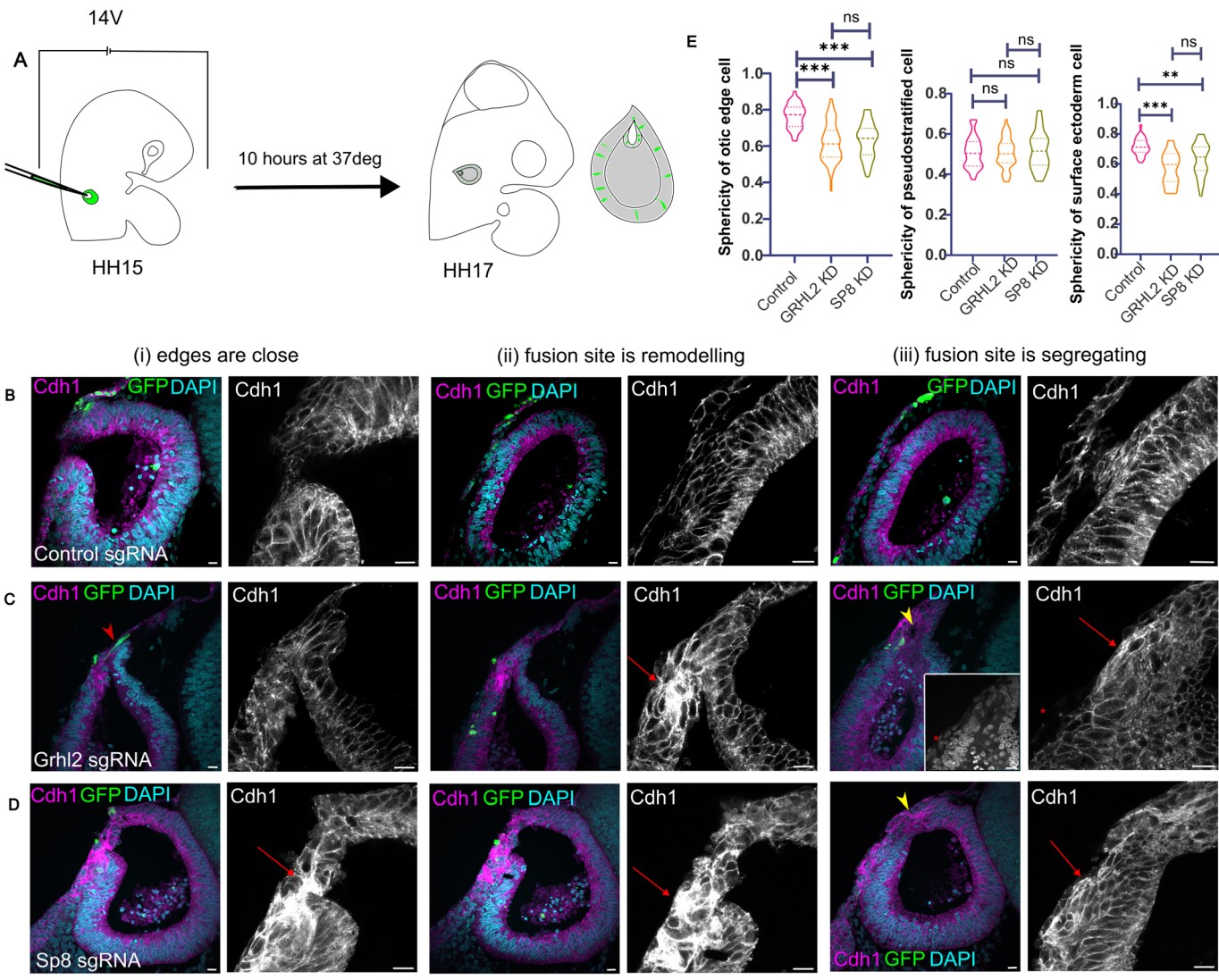

**Fig. 5. Genetic perturbation of epithelial fusion.** Different stages of epithelial fusion are depicted here: (i) edges are close, (ii) site of fusion is remodelling, (iii) fusion site is segregating. Sections are stained with DAPI (cyan), CDH1 (magenta) and GFP (green) in (A) Embryos electroporated with control-sgRNA (B) Embryos electroporated with Grhl2-sgRNA. The red arrowhead marks a GFP positive cell that is present at the site of fusion, the red arrow marks the accumulation of Cdh1 in the edge cells and the red asterisk marks the loss of Cdh1 in the surface ectoderm cells. The yellow arrowhead marks the ectoderm that has not segregated yet. (C) Embryos electroporated with Sp8-sgRNA. The red arrow marks the accumulation of Cdh1 in the edge cells and the red asterisk marks the loss of Cdh1 in the surface ectoderm cells. The yellow arrowhead marks the ectoderm that has not segregated yet; $n=5$ embryos. (E) Sphericity of OE, PS and SE cells in electroporated embryos. The number of cells for each condition for: OE, control gRNA $n=37$; Grhl2 gRNA $n=96$, SP8 gRNA $n=57$; PS, control gRNA $n=45$, Grhl2 gRNA $n=44$, SP8 gRNA $n=21$; and SE, control gRNA $n=28$, Grhl2 gRNA $n=11$, SP8 gRNA $n=30$. Significance tested by one-way ANOVA. Scale bars: 10 μm.

necessary for the round OE cells to integrate into either the OV or SE. Such intercalation may be driven by junctional remodelling. The diffuse localisation of junctional proteins in round cells may regulate stiffness and allow a more fluid and labile cell state, allowing junctions with either SE or OV neighbours to readily form.

Rac1 plays a central role in coordinating polarity and ECM assembly. Its expression, along with F-actin, was detected in interstitial cells that also had laminin punctae around them (Fig. 3E), suggesting it orchestrates actin-mediated basal matrix organisation (O'Brien et al., 2001). Notably, Rac1 is expressed in premigratory cell populations (Kee et al., 2007). Its downregulation by the proteoglycan, Syndecan4 is necessary for initiating migration (Matthews et al., 2008). The close homologue Rac3 is specifically expressed in the otic edge (Fig. 4L), further suggesting a role for Rac family members in regulating the cell state transitions required for fusion.

Transcriptomic analysis of the OE region revealed differential expression of genes involved in ECM remodelling, gastrulation, and Wnt signalling (Tables S1 and S2), processes also enriched during optic fissure fusion (Hardy et al., 2019). However, interpreting chick bulk RNA-seq data remains challenging due to limited genome annotation and gene ontology mapping resources. Functional dissection of candidate genes required targeted knockdown, allowing us to distinguish roles in differentiation versus epithelial fusion.

We focused on two transcription factors with opposing roles: Grhl2, an epithelial maintenance factor, and Sp8, an EMT inducer. *Grhl2* zebrafish mutants exhibit enlarged OVs and malformed semicircular canals, structures arising from the dorsal OV where fusion occurs (Han et al., 2011). *Sp8* mutants in *Xenopus* display broad defects in the auditory and vestibular systems (Chung et al.,

2014). Mosaic knockdown of either gene led to CDH1 accumulation in cells at the fusion interface and a loss of round cells, replaced by pseudostratified or squamous morphologies (Fig. 5). These data suggest that proper junction turnover and partial EMT are disrupted under knockdown conditions, resulting in impaired force transmission and cell shape change.

Accumulation of CDH1 likely increases cortical stiffness, reducing the ability of cells to undergo necessary morphological transitions. This is supported by prior studies showing that Grhl2 overexpression stiffens epithelia and impedes morphogenetic movements (Nikolopoulou et al., 2019). Interestingly, despite the mosaic nature of our knockdown, fusion was disrupted tissue-wide. This non-cell-autonomous effect may be mediated through either chemical signalling or mechanical coupling of junctional tension across epithelial sheets. Importantly, the epithelia did not segregate. This could be attributed to a lack of a fluidic state of the round cells, which we postulate are required for proper epithelial fusion. Proper localisation of junctions is important to accommodate cell shape changes without compromising their function (Levayer and Lecuit, 2013; Roeth et al., 2009). Thus, it is possible that Grhl2 and Sp8 are key regulators of the partial EM phenotype of OE round cells, and by regulating CDH1 localisation, they regulate epithelial fusion. Future studies should explore the additional candidates identified in our transcriptome, such as *Mlc9*, *Exoc4*, *Rac3* and *β-catenin*, to unravel the broader gene regulatory network underlying otic fusion. Additionally, live imaging and biophysical measurements of junctional tension could provide deeper insight into the emergent behaviours driving epithelial fusion and segregation during inner ear development.

## MATERIALS AND METHODS

### Chicken eggs
Fertilized *Kaveri* chicken eggs were obtained from the Central Poultry Development Organisation and Training Institute, Hessaraghatta, India. Eggs were stored at 25°C until use and subsequently incubated at 37°C in a humidified incubator. Embryos were staged according to the Hamburger and Hamilton (1951) developmental staging system.

### Immunohistochemistry (IHC)
Cryosections were washed three times in 0.1% Tween-20 in phosphate buffered saline (PBST) for 15 min each. Blocking was performed at room temperature (RT) for 1 h in PBST containing 3% heat-inactivated goat serum and 2 mg/ml bovine serum albumin (BSA). Sections were then incubated with primary antibodies (see Table 1 for details and dilutions), diluted in the blocking solution, for approximately 36–40 h at 4°C.

Following primary incubation, sections were washed three times in PBST (15 min each), and re-blocked for 1 h at RT. Secondary antibodies and fluorophore-conjugated phalloidin (see Table 1 for details and dilutions) were diluted in blocking solution and applied to the sections for 3 h at RT. After incubation, sections were washed again three times in PBST for 15 min each.

Nuclei were counterstained with DAPI (7–10 min), followed by a final wash in PBS for 10 min. Coverslips were mounted using Fluoroshield mounting medium. Images were acquired using an Olympus FV3000 confocal microscope equipped with high-sensitivity detectors, using 10× and 63× objectives, at a resolution of 2048×2048 pixels.

### SEM
Embryos were accessed using paper windows and immediately fixed in primary fixative containing 2.5% glutaraldehyde and 2% paraformaldehyde in 0.1 M sodium cacodylate buffer at 4°C for 24 h. Following fixation, embryos were washed three times in 0.1 M sodium cacodylate buffer on ice. Post-fixation was performed in 1% osmium tetroxide ($OsO_4$), chilled on ice, for 2 h. To eliminate residual yolk and paper fibres, embryos were rinsed three times with filtered Milli-Q water. Dehydration was initiated with two washes in 50% ethanol for 20 min each, followed by a graded ethanol series (70%, 90%, 95%, 99.5% and 100%) for 15 min per step. Final dehydration involved two 20-min washes in 100% absolute ethanol.

Dehydrated embryos were subjected to critical point drying using a Leica EM CPD300, following the standard 'sludge worm' protocol. Dried specimens were sputter-coated with a 10 nm gold layer using an Emitech K550X sputter coater. Imaging was performed on a Zeiss Merlin Compact VP SEM system.

For imaging the cellular morphology within the OV, embryos were carefully bisected through the OV using a scalpel prior to primary fixation.

### Imaris-surface module
Transverse cryosections from HH16 and HH17 stage chick embryos were stained with fluorescently labelled phalloidin to visualise cortical F-actin, which demarcates the approximate cell boundaries. Z-stacks were acquired from these sections and processed using the Imaris (Bitplane) surface tool to reconstruct the three-dimensional boundaries of individual cells of interest manually. Cell outlines were traced across the z-stacks to generate surface-rendered objects.

**Table 1. List of antibodies and their dilutions**

| Primary antibodies | Catalogue number | RRID | Dilution used |
|---|---|---|---|
| Rac1 ms | BD biosciences 610650 | AB_397977 | 1:100 |
| E cadherin ms | BD biosciences 610182 | AB_397581 | 1:100 |
| ZO1 ms | Thermofisher 33-9100 | AB_2663169 | 1:100 |
| ZO1 rbt | Thermofisher 40-2300 | AB_2533457 | 1:100 |
| Connexin 43 rbt | Sigma C6219 | AB_476857 | 1:100 |
| Ezrin ms | Sigma E8897 | AB_476955 | 1:100 |
| Laminin rbt | Sigma L9393 | AB_477163 | 1:100 |
| Pax2 rbt | Thermofisher 71-6000 | AB_2533990 | 1:100 |
| Anti GFP ms | Roche 11814460001 | AB_390913 | 1:100 |
| Anti GFP rbt | Abcam ab290 | AB_303395 | 1:100 |
| Caspase 3 Recombinant Rabbit Monoclonal Antibody (9H19L2) | Invitrogen 700182 | AB_2532293 | 1:100 |
| Anti-Digoxigenin | Roche 11093274910 | AB_514497 | 1:2000 |
| Secondary Antibody and Dyes | | | |
| Goat Anti-mouse IgG (H+L) Alexa Flour Plus 555 | Invitrogen A32727 | AB_2633276 | 1:1000 |
| Alexa Fluor® 488 goat anti-rabbit IgG (H+L) *2 mg/ml* | Invitrogen A11008 | AB_143165 | 1:1000 |
| Alexa Fluor® 647 goat anti-rabbit IgG (H+L) *highly cross-adsorbed* *2 mg/ml* | Invitrogen A21245 | AB_2535813 | 1:1000 |
| Goat anti-Mouse IgG (H+L) Cross-Adsorbed Secondary Antibody, Alexa Fluor™ 647 | Invitrogen A21235 | AB_2535804 | 1:1000 |
| Alexa Fluor 488 phalloidin | Invitrogen A12379 | | 1:400 |
| Molecular Probes, Alexa Fluor 568 phalloidin | Invitrogen A12380 | | 1:400 |
| DAPI | Sigma D9542 | | 1:2000 |

Three distinct epithelial cell populations were analysed: (1) pseudostratified cells within the OV, (2) rounded interfacial cells at the fusion edge and (3) squamous cells of the SE that are continuous with the OV epithelium. The shape descriptor 'sphericity' was used to distinguish these populations quantitatively. Statistical significance among the three groups was assessed using one-way analysis of variance (ANOVA).

### Protein localisation analysis using ImageJ

Sections of HH16 and HH17 stage chick embryos were immunostained for various proteins using the immunohistochemistry protocol described above. To quantify the spatial distribution of protein localisation at the cellular level, a single optical section from each z-stack was selected, and individual cell boundaries were manually traced using the freehand line tool in Fiji (ImageJ).

Protein intensity along the cell periphery was extracted using the 'plot profile' function. The coefficient of variation (CV) of pixel intensity values along the circumference was calculated for each cell. The CV, defined as the standard deviation divided by the mean intensity along the cell boundary, served as a metric to assess heterogeneity in protein distribution. A lower CV indicated a more uniform (diffuse) localisation, whereas a higher CV reflected a punctate or spatially restricted distribution. This method allowed us to quantitatively compare the expression patterns of proteins across the three epithelial cell types analysed: pseudostratified OV cells, rounded edge cells and squamous SE cells.

### Whole-mount *in-situ* hybridisation

All solutions used were treated with 0.1% diethyl pyrocarbonate (DEPC) to inactivate RNases. DEPC was added to a final concentration of 0.1% and incubated at 37°C overnight. Solutions and components that could not be autoclaved were prepared using DEPC-treated water to ensure RNase-free conditions.

Chick embryos were fixed in 4% paraformaldehyde (PFA) for 2 h at RT, followed by thorough washing in DEPC-treated PBS (DEPC-PBS) to remove residual fixative. The allantoic membrane was removed, and embryonic cavities were punctured to facilitate reagent penetration. Embryos were dehydrated through a graded methanol and PBS series (25%, 50%, 75%, 90%, 95% 99%) and washed twice with 100% methanol before storage at −20°C in 100% methanol.

For *in-situ* hybridisation, embryos were rehydrated gradually into PBST (PBS with 0.1% Tween-20) on ice. Proteinase K treatment was performed at a concentration of 10 µg/ml in DEPC water, with the incubation time corresponding to the embryo's Hamburger–Hamilton (HH) stage (e.g. HH17 embryos were treated for 17 min) on ice. Following digestion, embryos were washed twice with PBST for 5 min each on ice and post-fixed in 4% PFA with 0.2% glutaraldehyde in PBST for 20 min at RT. Three 5-min washes followed post-fixation in PBST. Embryos were then transferred into pre-hybridisation solution and allowed to sink before being incubated in fresh pre-hybridisation solution at 65°C for 1 h. Digoxigenin (DIG)-labelled RNA probes were added to a final concentration of 1 µg/ml, and hybridisation was carried out overnight at 65°C. Post-hybridisation washes included a series of high-stringency washes at 65°C, followed by aqueous washes in maleic acid buffer with

Tween-20 (MABT). Blocking was performed using the Roche Blocking Reagent according to the manufacturer's instructions. The blocking solution was supplemented with 20% heat-inactivated goat serum before incubation with anti-DIG-AP antibody (Roche) at a dilution of 1:2000, overnight at 4°C.

Embryos were washed extensively with MABT over a full day at RT to reduce non-specific staining. Colour was developed in NTMT buffer using NBT/BCIP substrate at RT without agitation. Alternatively, BM-Purple substrate (Roche) was used for colourimetric detection. After colour development, embryos were equilibrated in 60% glycerol for optical clearing and imaged using a stereomicroscope.

DIG-labelled RNA probes were synthesised using Roche reagents and primers listed in Table 2, following the manufacturer's protocol.

### Tissue dissection and RNA preparation

All dissection tools and Petri dishes were treated to eliminate RNase contamination. Tools were baked at 200°C for 5 h, and all surfaces, including microscope stages and workbenches, were cleaned thoroughly with RNaseZap (Thermo Fisher Scientific). PBS used during dissection was DEPC-treated and autoclaved before use. Dispase (2.4 U/ml; xxx supplier) was prepared in DEPC-treated PBS, aliquoted, and stored at −20°C. Dispase was used to remove mesenchymal cells adhering to the SE and OV; however, the otic edge tissue was not treated with dispase due to its small size and susceptibility to damage.

Embryo dissections were performed using electrolytically sharpened tungsten needles, which were frequently sterilised by passing through an open flame. Embryos were pinned to a Sylgard-coated dish using dissection pins to ensure stability during tissue isolation.

The otic edge was dissected first. A tungsten needle was used to perforate the area around the otic pore from the dorsal side, followed by perpendicular perforations to isolate the intact otic pore region completely. The dissected tissue was immediately transferred to a sterile 1.5 ml tube containing 1 ml of TRI reagent (Invitrogen, xxxcatalogue code) for RNA preservation.

Next, the OV (excluding the edge) was dissected by gently nudging the structure without damaging the surrounding tissue. The isolated OV was incubated in a droplet of dispase solution for 10 min in a sterile 35 mm dish to remove residual mesenchymal cells. Following digestion, the tissue was transferred to a 1.5 ml tube containing 1 ml of TRI reagent.

The SE was then dissected using a single tungsten needle to gently displace the tissue, followed by using two needles in a scissor-like manner to excise a small region of epithelium. The dissected tissue was incubated in dispase solution for 10 min and then transferred into a 1.5 ml tube containing TRI reagent (xxxxsupplier, catalogue code).

For each RNA sample, eight tissue pieces were pooled into a single tube and stored at −20°C until RNA extraction. Total RNA was isolated using the TRI reagent protocol, and the quality and concentration of RNA were assessed using 1 µl of the sample for each of the following: Bioanalyzer (Agilent) for RNA integrity and Qubit fluorometer (Invitrogen, xxxxcat code) for concentration measurement. RNA samples were resuspended in RNase-free water and stored at −80°C until cDNA library preparation.

**Table 2. List of primers used to generate whole-mount *in-situ* hybridisation antisense probes**

| Gene name | Forward primer | Reverse primer |
| --- | --- | --- |
| Grhl2 | GCCTGGAAGTCGTATTTGGA | GTCAAGGACCCTCTGCTTTG |
| Grhl3 | GGCATCCCTACTCCTCACAA | TATAGGAATTCAGTCTCTGGCCTCA |
| Snail2 | AGACAGATCCAATCTGAGGG | CTCTCTTGCACTTATTCCCG |
| Zeb2 | TTCCTGGTCCCATTCCGTTG | GGTCGCAACCCAGGAATACT |
| Dach1 | CTTCAGACAGAATCCCTGTCCA | CCACAGAACTCCATTTGGGTGA |
| Sp8 | GGCGCATTTGGATCATTCCC | GGCAAACCGCTAATGTGGTG |
| Msx1 | GAAGCAGTACCTGTCCATC | ATAGTACACAGAGAGAGCCC |
| Bambi | ATTGCTGTTCCTATAGCTGG | ACATAGGAGCTTGCATACAG |
| Rac3 | GCTGTAGGGAAGACCTGCTT | CAGGATGGATGTCTCAGCCC |
| Itgb3 | GTGGAGTGCAAGAAGTATGA | TGTGTCCTCAACACATAAGG |
| Wnt 3A | CGATTCTGTCGGAACTATGT | GGAGCCTTGAAGAAGTTGTA |
| EphA4 | AGCATTGCTTGATGTACAGA | GGCAAATTGAACTGCTTCTT |

## CRISPR–Cas9 construct

The control-sgRNA sequence 5-GCACTGCTACGATCTACACC-3 was adopted as previously used elsewhere and cloned into Bbs1-flanked cloning region of pCAG-SpCas9-GFP-U6-gRNA (Gandhi et al., 2017). The guide sequences for *Grhl2* and *SP8* were designed using CRISPOR webtool and were cloned into the aforementioned vector (Concordet and Haeussler, 2018). The Grhl2-sgRNA used was 5-TGGCAGCTCCACGCCAATTACGG-3 and SP8-sgRNA was 5-CACGCAGTCACGTCGGCGAGCGG-3.

## CRISPR–Cas9 electroporation

Fertilised *Kaveri* chicken eggs were incubated for 50 h to obtain Hamburger–Hamilton stage 15 (HH15) embryos (Fig. 2.2). DNA injection and electroporation were performed directly into the developing OV.

Microcapillary glass needles were loaded with the DNA injection mix consisting of 5 µg/µl plasmid DNA (xxsupplier, cat codexx), 30% sucrose (xxsupplier, cat codex) and 0.05% Fast Green (xxsupplier, cat codex) to visualise the injection volume. The vitelline membrane over the embryo was carefully removed, and the DNA mix was gently injected into the lumen of the OV using a micromanipulator, avoiding physical contact with the embryonic tissue (Fig. 2A).

Electroporation was carried out immediately after injection using platinum electrodes on either side of the OV. Five square-wave pulses (14 V, 50 ms duration, 100 ms intervals) were applied using a square-pulse electroporator (xxsupplier). Following electroporation, embryos were cultured for 8 h in albumin agar plates at 37°C before fixation or further processing.

## Statistical analysis

Ordinary one-way ANOVA was performed to test significance of data in the paper. Tukey's multiple comparisons test was performed with an alpha value of 0.05. These tests were performed using GraphPad Prism software.

## Acknowledgements

This work was supported by the Department of Atomic Energy, Government of India, (RTI 4006), and grants from ANRF-SERB (CRG/2018/001235), the Royal National Institute for Deaf People (IPG programme) and TIFR Infosys-Leading Edge Grant. We acknowledge the support of the Electron Microscopy Facility and Central Imaging Facility at NCBS. We thank Rolf Eriksson for his valuable inputs in bulk mRNA sequencing. We also thank Dr Awadesh of the NGS facility in NCBS, for performing the mRNA sequencing. Shivangi Pandey and Nidhi Parikh helped in the standardisation of the *in-situ* hybridisation protocol. Raman Kaushik helped in the design of the CRISPR-gRNA. We thank CSIR for JRF-SRF Fellowship to V.N.T.

## Competing interests

The authors declare no competing or financial interests.

## Author contributions

Conceptualization: R.K.L.; Data curation: V.N.T., R.K.L.; Formal analysis: V.N.T., R.K.L.; Funding acquisition: R.K.L.; Investigation: V.N.T., H.P.; Methodology: V.N.T., H.P.; Project administration: R.K.L.; Supervision: R.K.L.; Validation: H.P.; Writing – original draft: V.N.T.; Writing – review & editing: R.K.L.

## Funding

 Deposited in PMC for immediate release.

## Data and resource availability

RNA-Seq data has been deposited in the NCBI Gene Expression Omnibus database with the accession number GSE297407. All other relevant data and details of resources can be found within the article and its supplementary information.

## Peer review history

The peer review history is available online at https://journals.biologists.com/bio/lookup/doi/10.1242/bio.062213.reviewer-comments.pdf

## First Person

This article has an associated First Person interview with the first author of the paper.

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
