## [Peer Review File · Biology Open]

Epithelial fusion is mediated by a partial epithelial-mesenchymal transition

Varsha N. Tamilkumar, Harsha Purushothama and Raj K. Ladher
DOI: 10.1242/bio.062213

Editor: Tristan Rodríguez

Review timeline

Original submission:	20 May 2025
Editorial decision:	1 July 2025
First revision received:	26 August 2025
Accepted:	26 August 2025

Original submission

First decision letter

MS TITLE: Epithelial fusion is mediated by a partial epithelial-mesenchymal transition

AUTHORS: Varsha N Tamilkumar; Harsha Purushothama; Raj K Ladher

Dear Raj,

I have now received all the referees' reports on the above manuscript, and have reached a decision. I am sorry to say that the outcome is not a positive one. The referees' comments are appended below, or you can access them online: please go to .

As you will see, the referees raise some significant concerns about your paper, and are not strongly in favour of publication. They are somewhat concerned both with the extent of advance of the findings and both have issues with the strength of the evidence in support of the claims. Having looked at the manuscript myself, I tend to agree with their views, and I must therefore, reject your paper.

I do realise this is disappointing news, but the journal receives many more papers than we can publish, and we can only accept manuscripts that receive strong support from referees.

I do hope you find the comments of the referees helpful, and that this decision will not dissuade you from considering the journal for publication of your future work. Many thanks for sending your manuscript to us.

Reviewer 1: SUMMARY OF THE ADVANCE MADE IN THIS PAPER AND ITS POTENTIAL SIGNIFICANCE TO THE FIELD

This paper examines epithelial fusion during development of the chick otic vesicle. A population of otic edge cells is identified that appear to undergo partial epithelium-to-mesenchyme (EMT) transition. A number of genes and cellular features that correlate with this population are described. Transcriptomics shows these cells to differ in gene expression from the surface ectoderm and otic vesicle cell populations. The experimental part of the study is electroporation to knock down Grhl2 and Sp8, genes whose expression appears regulated during epithelial fusion. The

findings are interpreted to show an adverse effect of knockdown on cellular and molecular aspects of otic development, but crucial controls are omitted from these studies.

Overall, this paper extends to otic development what has previously been extensively identified and investigated in many other morphogenetic fusion events: e.g. palatal shelves, neural folds, optic fissure, etc. I am not convinced that this paper moves our understanding of epithelial fusion forwards significantly, beyond these previously studies.

SUGGESTIONS TO AUTHORS

There are no page numbers on the manuscript which made reviewing difficult. In the Results section:

P 1, first para. "Basal lamina" is referred to here and elsewhere in the manuscript. In the legend to Fig 1G this is in relation to laminin staining. However, the basal lamina is a structure visible only at transmission electron microscopic level. The 'basement membrane' (not 'basal membrane' as in Fig 3E legend) is shown by laminin staining, and this term should be used.

P 1, second para, phalloidin staining is referred to, but this is not mentioned in the legend to Fig 1.

P 3, third para. Laminin is said to change from fibrillar to diffuse and then back to fibrillar (Fig 3E) but this is not clear - Fig 3E is not well labelled and this makes it hard to see what the authors are claiming.

P 7. The experimental part of this study involved electroporation of CRISPR-Cas9-mediated mosaic knockdowns. GFP is taken as an indication of electroporated cells, but we are not shown any evidence that *Grhl2* or *Sp8* are knocked down in electroporated cells. Moreover, GFP+ve cells appear very sparse indeed. Accumulation of *Cdh1* expression is seen in the middle panels of Fig 5C, D, but this does not seem to coincide with GFP expression, so can it be attributed to the alteration in target gene expression? Indeed, the authors acknowledge this: "Although the CRISPR knockdowns were mosaic, the phenotypes observed were tissue-wide", but their explanation, which invokes long distance biomechanical effects of localized *Cdh1* perturbation is not convincing, especially given the authors' finding that "the pseudostratified morphology of cells mediating fusion remained largely unperturbed". The authors say there is loss of tissue segregation in *Grhl2* and *Sp8* sgRNA electroporations (right panels), but could this be simply due to non-specifically delayed otic development?

Fig 1A is not cited in the text.

Fig 2 and legend. It would be helpful to show the name of the protein on each of the graphs B, D, F. The insets in parts A, C, E, G are not mentioned in the legend.

Fig 3. The red arrows in A-C are very small and hard to see in some cases. Parts D-H have multiple images but these are not well described in the legend. In part F, "... seem to be secreting laminin" is pure conjecture and should be removed. Similarly in part G, "... a diffused expression of *Cdh1* at the site of remodelling, as marked by diffused laminin expression" is hard to understand. Do the authors mean to use the word 'diffuse' (i.e. spread over a wide area), and how can laminin distribution mark *Cdh1* expression? In part H, is the name *Pax2* or *pax2*?

Fig 4. Part D is hard to understand. What do the letters A-D mean: e.g. SE.A, SE.B, etc. Are these biological replicates of the assay? This needs to be made clear in the legend. Please re-define DEG here for clarity. Part E appears to be a repeat of Fig 1A. Can it be removed?

Reviewer 2: SUMMARY OF THE ADVANCE MADE IN THIS PAPER AND ITS POTENTIAL SIGNIFICANCE TO THE FIELD

This manuscript investigates epithelial fusion during otic vesicle closure in chick embryos. The authors identify a distinct population of cell at the otic edge (OE). Using a combination of morphological analysis, immunostaining, and transcriptomic profiling, they show that OE cells

exhibit reduced apicobasal polarity, junctional remodeling, high sphericity, and altered expression of ZO-1, CDH1, and RAC1. Transcriptomic data show an enrichment of EMT regulators and WNT signaling components. Knockdown of transcription factors Grhl2 and Sp8 affects the localization of CDH1. They present a model where OE cells are thought to be in a partial epithelial-to-mesenchymal transition (EMT) state, balancing junctional remodeling and tissue cohesion during epithelial fusion. Although the manuscript is clearly written, the conclusions are currently undermined by the reliance on low-resolution imaging. Higher-resolution imaging data, along with more mechanistic insights and a well-documented statistical analysis, are crucial for substantiating the claims of partial EMT.

SUGGESTIONS TO AUTHORS

General

- 1) Please add line numbers to the text for review purposes
- 2) Most images are intended to display subcellular localization. However, with the magnification and resolution chosen (and the compression of image files), it's impossible to see the localization of various proteins.
- 3) Please mention the n for every image (how many biological replicates were used for each experiment) and quantifications
- 4) Please report all statistics clearly in the methods section.
- 5) Please consider perturbing CDH1 to test the molecular and cellular mechanism of partial EMT. I expect it to affect RAC1, ECM, and cell morphologies.
- 6) Please consider examining CDH2 to test the partial EMT model further.

Introduction

- 1) The role of EMT during fusion is not novel: <https://pmc.ncbi.nlm.nih.gov/articles/PMC4523936/>
- 2) "This process converts what was once a continuous epithelial sheet into two distinct epithelia"

This is not necessarily true, as epithelial fusion does not always convert a continuous epithelial sheet into distinct epithelia. For example, during both sprouting and intussusceptive angiogenesis, fusion does not create two distinct epithelia.

Related to Figure 1

- 1) Why is sphericity important is not explained. Rounded cells could also mean cells on their way to division. Is this a proliferative tissue?
- 2) Please clearly mark PS, OE and SE in the figure itself
- 3) "Cells with a sphericity index close to 1 were identified in the otic edge cells"

What sphericity index value was set as the threshold in figure 11? Please label it "population of "round cell".

Related to Figure 2

- 1) I appreciate the CV measurement, but where the ROI was taken is not shown. Please acquire/show better quality images for this figure as it is critical for all the conclusions drawn in the manuscript.
- 2) Rac1 has several functions both in epithelial and migratory cells. Not sure if CV is the right measurement if Rac1 is absent.
- 3) "Using immunostaining for laminin, we found that a proportion of round cells do not contact the basal lamina. In contrast, SE and OV cells remain in contact with the basal lamina (Fig 2G)". Quantification of this proportion was not provided. Please quantify the proportion across different biological replicates and plot it.

4) 2H: I am not convinced those are filopodia.

Related to Figure 3

1) "Later, we observe apoptosis in only some of the interstitial cells in HH17+(Fig 3D). This may suggest that interstitial cells have re-integrated into either the SE or OV"

The authors observe a population of transient interstitial cells that appear after fusion; the way this observation is written implies that the interstitial cells are SE/OV-derived cells (possibly OE) that leave the epithelia after fusion, especially as they suggest that the cells "re-integrate" into the SE/OV. This claim is too strong, as there is no evidence for the identity of the interstitial cells. They may be SE/OV derived, but they may also be mesenchymal cells that migrated from surrounding tissues. This can be resolved with lineage tracing.

The decrease in apoptosis does not imply re-integration. It's possible that some interstitial cells migrated away from the fusion site.

2) "Notably, some interstitial cells secrete laminin and exhibit polarised Rac1 expression, suggesting a role in orienting neighbouring cells (Fig 3F inset)"

The presence of laminin around a cell does not necessarily indicate that that specific cell secretes laminin; laminin can be deposited by other cells.

3) Please cite a paper that supports the claim that Rac1 expression in one cell affects neighboring cell polarity.

4) "To understand whether edge cells can switch lineage, we examined the otic fate marker Pax2 [...] We found that it is transiently expressed during fusion in the edge cells during fusion, but is completely absent from the site following segregation (Fig 3I)"

The loss of Pax2 in the otocyst after fusion could be due to the loss of OE cells via various possible mechanisms (apoptosis, necrosis, cell extrusion, etc.) and not a change in OE cell identity.

To make a claim about lineage, the authors should conduct a lineage tracing experiment and show that OE cells undergo different states.

5) "These findings suggest that a coordinated interplay between polarity proteins, ECM remodelling, and junctional complexes drives cell sorting during otic vesicle closure."

The authors have characterized the localization of various proteins but have not mechanistically shown that these proteins are what drives cell sorting. To make this claim, the authors should perturb these processes and report on how they affect cell sorting and fusion.

6) 3D: Please add a dashed line that indicates the boundary between the otocyst, epidermis, and interstitial space between them.

7) 3E: Please annotate images with stages

Related to Figure 5

"Overall, we observed a spatially restricted and balanced expression of transcription factors and signalling molecules that either promote or inhibit WNT, TGF β , and BMP signalling. This balance likely sustains the partial EMT state of OE cells."

What does "balanced" mean? How does one judge if transcription factor expression is "balanced" between promoters and inhibitors, when each transcription factor is expressed at different levels and can work through different mechanisms?

First revision

Author response to reviewers' comments

Reviewer 1

1. P 1, first para. "Basal lamina" is referred to here and elsewhere in the manuscript. In the legend to Fig 1G this is in relation to laminin staining. However, the basal lamina is a structure visible only at transmission electron microscopic level. The 'basement membrane' (not 'basal membrane' as in Fig 3E legend) is shown by laminin staining, and this term should be used.

This has now been updated in the manuscript

2. P 1, second para, phalloidin staining is referred to, but this is not mentioned in the legend to Fig 1.

We now include reference to phalloidin, which was used to mark the outlines of cells,

3. P 3, third para. Laminin is said to change from fibrillar to diffuse and then back to fibrillar (Fig 3E) but this is not clear - Fig 3E is not well labelled and this makes it hard to see what the authors are claiming.

In Fig 3E, the otic vesicle is contiguous with the surface ectoderm when the edges are apart and the laminin staining is uniform and fibrillar. After the closure of the otic vesicle, the surface ectoderm and the otic vesicle are segregated into two separate epithelia with uniform laminin staining. However, when the edges are fusing, laminin is sparsely beginning to localise at this site of fusion, thus enabling the segregation of otic vesicle from the surface ectoderm. This sparse localisation is referred to as "diffused" in contrast to the fibrillar pattern before and after fusion. We have revised the terminology in the manuscript, replacing 'fibrillar' with 'robust' and 'diffuse' with 'sparse'. These changes make the description clearer for the reader.

4. P 7. The experimental part of this study involved electroporation of CRISPR- Cas9-mediated mosaic knockdowns. GFP is taken as an indication of electroporated cells, but we are not shown any evidence that *Grhl2* or *Sp8* are knocked down in electroporated cells. Moreover, GFP+ve cells appear very sparse indeed. Accumulation of *Cdh1* expression is seen in the middle panels of Fig 5C, D, but this does not seem to coincide with GFP expression, so can it be attributed to the alteration in target gene expression? Indeed, the authors acknowledge this: "Although the CRISPR knockdowns were mosaic, the phenotypes observed were tissue-wide", but their explanation, which invokes long distance biomechanical effects of localized *Cdh1* perturbation is not convincing, especially given the authors' finding that "the pseudostratified morphology of cells mediating fusion remained largely unperturbed". The authors say there is loss of tissue segregation in *Grhl2* and *Sp8* sgRNA electroporations (right panels), but could this be simply due to non-specifically delayed otic development?

Unfortunately, the antibodies available for *Grhl2* and *SP8* did not give convincing staining in the chick. Moreover, as both constructs only generated indels in *Grhl2* and *SP8* coding sequence, these alterations could not be detected using *in situ* hybridisation. The effectiveness of the gRNA is known (through an endonuclease assay in DF1 cells). Both genes are only expressed in the otic edge (Fig 4I and 4M) and so their knockdown is expected to only effect these cells. *Grhl2* is known to regulate *Cdh1* expression (Werth et al. 2010). We observe down-regulation of *Cdh1* in electroporated cells. We noted that the characteristic shape of the edge cells in the wild type was perturbed in the case of CRISPR-Cas9 mediated mosaic knockdowns. This perturbation was not seen in control gRNA electroporations. To verify whether there was a delay in development we cultured the embryos for 24 hours after electroporation. These embryos display signs of an inner ear

characteristic of these stages, with the only difference that the edges had not fused and remain contiguous with the non- neural ectoderm. We have rewritten this section to make our controls and observations clearer.

5. Fig 1A is not cited in the text.

This is now cited.

6. Fig 2 and legend. It would be helpful to show the name of the protein on each of the graphs B, D, F. The insets in parts A, C, E, G are not mentioned in the legend.

This is now resolved.

7. Fig 3. The red arrows in A-C are very small and hard to see in some cases. Parts D-H have multiple images but these are not well described in the legend. In part F, "... seem to be secreting laminin" is pure conjecture and should be removed. Similarly in part G, "... a diffused expression of Cdh1 at the site of remodelling, as marked by diffused laminin expression" is hard to understand. Do the authors mean to use the word 'diffuse' (i.e. spread over a wide area), and how can laminin distribution mark Cdh1 expression? In part H, is the name Pax2 or pax2?

We thank the reviewer for pointing this out. We have increased the arrow sizes and added more information in the legends. We have also removed the conjecture, and tightened up our language, as well as consistency in labelling genes.

8. Fig 4. Part D is hard to understand. What do the letters A-D mean: e.g. SE.A, SE.B, etc. Are these biological replicates of the assay? This needs to be made clear in the legend. Please re-define DEG here for clarity. Part E appears to be a repeat of Fig 1A. Can it be removed?

We thank the reviewer for pointing this out. We have re-presented this figure, and taken greater care in our definitions.

Part D is the heatmap generated with the Differential Gene Expression (DGE) analysis. As part of this, two separate comparisons were made: transcriptome of otic edge (OE) versus otic vesicle (OV) and transcriptome of otic edge (OE) versus surface ectoderm (SE). The analyses produced a list of genes differentially expressed between the two sample types. Top 30 genes from this list were used to generate the heatmaps as seen in Part D. The letters A-D represent the biological quadruplicates of the corresponding tissue type.

Reviewer 2

1. Please add line numbers to the text for review purposes

These are added

2. Most images are intended to display subcellular localization. However, with the magnification and resolution chosen (and the compression of image files), it's impossible to see the localization of various proteins.

We have remade the figures so that the images are not subject to compression artifacts.

3. Please mention the n for every image (how many biological replicates were used for each experiment) and quantifications

This is added in the legends

4. Please report all statistics clearly in the methods section.

This is added in the methods

5. Please consider perturbing CDH1 to test the molecular and cellular mechanism of partial EMT. I expect it to affect RAC1, ECM, and cell morphologies.

Cdh1 is expressed throughout the surface ectoderm and the otic vesicle. Perturbing it will have effects that will make it harder to determine specifically its role in the edge cell population.

6. Please consider examining CDH2 to test the partial EMT model further.

We had tested Cdh2 and other atypical Cadherins present during neural tube closure and neural crest cell migration, but they were not present in the otic edge cell population. We observed an altered localisation of Cdh1 in edge cells. Upon genetic perturbation of Grhl2 and SP8, we observed mis-regulated Cdh1 localisation.

Introduction

7. The role of EMT during fusion is not novel:

<https://pmc.ncbi.nlm.nih.gov/articles/PMC4523936/>.

We thank the reviewer for this comment. While the role of EMT in fusion is not new, the implication that the fusing cells are undergoing a partial EMT (or hybrid EMT) is. This is interesting in the otic vesicle where those cells that are fusing are not meant to emigrate (and so will not exhibit the potentially confounding effects of EMT during neural crest formation. We have motivated this study and our findings more clearly in the revised manuscript.

8. "This process converts what was once a continuous epithelial sheet into two distinct epithelia" This is not necessarily true, as epithelial fusion does not always convert a continuous epithelial sheet into distinct epithelia. For example, during both sprouting and intussusceptive angiogenesis, fusion does not create two distinct epithelia. Related to Figure 1

We thank the reviewer for this insight. We have rewritten this section.

9. Why is sphericity important is not explained. Rounded cells could also mean cells on their way to division. Is this a proliferative tissue?

We have described sphericity better and articulated the reason why we are using it more clearly. Rounded cells can indeed be proliferative - and this is particularly true for pseudostratified epithelia, where mitosis occurs apically. The edge cell population has some cells that are dividing (as observed using PH3 staining); however, the proportion is no more than in the rest of the epithelium.

10. Please clearly mark PS, OE and SE in the figure itself

The cells are marked.

11. "Cells with a sphericity index close to 1 were identified in the otic edge cells" What sphericity index value was set as the threshold in figure 11? Please label it "population of "round cell".

We counted the cells in the edge manually. Automated thresholding to distinguish the cells in the otic vesicle was not possible. We used confocal images of cryo-sections to count the cells. Based on cell shape across multiple optical stacks, the cells were designated as round cell vs pseudostratified cell.

Related to Figure 2

12. I appreciate the CV measurement, but where the ROI was taken is not shown. Please acquire/show better quality images for this figure as it is critical for all the conclusions drawn in the manuscript.
This has now been rectified.

13. Rac1 has several functions both in epithelial and migratory cells. Not sure if CV is the right measurement if Rac1 is absent.

That is correct. We will represent the absence of expression in a different way. The primary reason is to emphasise that there are three populations of cells with different Rac1 localisation.

14. "Using immunostaining for laminin, we found that a proportion of round cells do not contact the basal lamina. In contrast, SE and OV cells remain in contact with the basal lamina (Fig 2G)". Quantification of this proportion was not provided. Please quantify the proportion across different biological replicates and plot it.

This quantification is provided in Fig 1I.

15. 2H: I am not convinced those are filopodia.

We now provided a better quality image of these, stained with Rac1, which is typically associated with filopodia.

Related to Figure 3

16. 1) "Later, we observe apoptosis in only some of the interstitial cells in HH17+(Fig 3D). This may suggest that interstitial cells have re-integrated into either the SE or OV" The authors observe a population of transient interstitial cells that appear after fusion; the way this observation is written implies that the interstitial cells are SE/OV-derived cells (possibly OE) that leave the epithelia after fusion, especially as they suggest that the cells "re-integrate" into the SE/OV. This claim is too strong, as there is no evidence for the identity of the interstitial cells. They may be SE/OV derived, but they may also be mesenchymal cells that migrated from surrounding tissues. This can be resolved with lineage tracing.
The decrease in apoptosis does not imply re-integration. It's possible that some interstitial cells migrated away from the fusion site.

The reviewer is correct. We reduce the strength of the claim in the revised manuscript.

17. 2) "Notably, some interstitial cells secrete laminin and exhibit polarised Rac1 expression, suggesting a role in orienting neighbouring cells (Fig 3F inset)" The presence of laminin around a cell does not necessarily indicate that that specific cell secretes laminin; laminin can be deposited by other cells.

We thank the reviewer for this comment. We have removed this statement in the revised manuscript

18. 3) Please cite a paper that supports the claim that Rac1 expression in one cell affects neighboring cell polarity.

This article also makes the claim that Rac1 can affect the polarity in a neighbouring cell - Jain, S., Cachoux, V.M.L., Narayana, G.H.N.S. et al. The role of single-cell mechanical behaviour and polarity in driving collective cell migration. Nat. Phys. 16, 802-809 (2020). <https://doi.org/10.1038/s41567-020-0875-z>

This study elucidated the role of a single polarised cell in collective epithelial cell migration. MDCK cells were cultured on microprinted plates, constrained by annular rings. This ensured the formation of a one-dimensional migrating cell track, which rotated either clockwise or counterclockwise. When two cell tracks moving in opposite directions collide, the leading cell lamellipodia of the track with a larger number of cells and higher speed repolarise the other leading cell, eventually changing its polarity. Using a fluorescent biosensor of Rac1 and Cdc42, they observed this repolarisation upon contact in real time. The lamellipodia of the winning cell extended below the lamellipodia of the other cell and repolarised it. When Rac1 gradients in the cells were perturbed by optogenetics for a whole cell activation of Rac1, random extensions of lamellipodia were generated and thus stopped the directed migration of cells.

19. 4) *"To understand whether edge cells can switch lineage, we examined the otic fate marker Pax2 [...] We found that it is transiently expressed during fusion in the edge cells during fusion, but is completely absent from the site following segregation (Fig 3l)" The loss of Pax2 in the otocyst after fusion could be due to the loss of OE cells via various possible mechanisms (apoptosis, necrosis, cell extrusion, etc.) and not a change in OE cell identity.*

To make a claim about lineage, the authors should conduct a lineage tracing experiment and show that OE cells undergo different states.

We thank the reviewer for this comment. We appreciate the perspective and have removed the claim on lineage based on the changing expression pattern.

20. 5) *"These findings suggest that a coordinated interplay between polarity proteins, ECM remodelling, and junctional complexes drives cell sorting during otic vesicle closure."*

The authors have characterized the localization of various proteins but have not mechanistically shown that these proteins are what drives cell sorting. To make this claim, the authors should perturb these processes and report on how they affect cell sorting and fusion.

We have removed the reference to cell sorting here.

21. 6) 3D: *Please add a dashed line that indicates the boundary between the otocyst, epidermis, and interstitial space between them.*

7) 3E: *Please annotate images with stages*

This is now added, and annotated.

22. *Related to Figure 5*

"Overall, we observed a spatially restricted and balanced expression of transcription factors and signalling molecules that either promote or inhibit WNT, TGFB, and BMP signalling. This balance likely sustains the partial EMT state of OE cells." What does "balanced" mean? How does one judge if transcription factor expression is "balanced" between promoters and inhibitors, when each transcription factor is expressed at different levels and can work through different mechanisms?

The reviewer is correct - this section of the discussion has been rewritten with a clearer perspective on the data in the manuscript.

Second decision letter

MS ID#: bio.062213R1

MS TITLE: Epithelial fusion is mediated by a partial epithelial-mesenchymal transition

AUTHORS: Varsha N Tamilkumar; Harsha Purushothama; Raj K Ladher

I am very happy to tell you that your manuscript has been accepted for publication in Biology Open, pending our standard publication integrity checks. It was accepted on 26th August 2025. Thank you for submitting to BiO!